METG-RESEARCH ARTICLE

# Most researchers would receive more recognition if assessed by article-level metrics than by journal-level metrics

**Salsabil Arabi, Chaoqun Ni, B. Ian Hutchins** ⓘ *

Information School, School of Computer, Data, and Information Sciences, College of Letters & Science, University of Wisconsin–Madison, Madison, Wisconsin, United States of America

* bihutchins@wisc.edu

## Abstract

During career advancement and funding allocation decisions in biomedicine, reviewers have traditionally depended on journal-level measures of scientific influence like the impact factor. Prestigious journals reject large quantities of papers, many of which may be meritorious. It is possible that this process could create a system whereby some influential articles are prospectively identified and recognized by journal brands, but most influential articles are overlooked. Here, we measure the degree to which journal prestige hierarchies capture or overlook influential science. We quantify the fraction of scientists' articles that would receive recognition because (a) they are published in journals above a chosen impact factor threshold, or (b) they are at least as well-cited as articles appearing in such journals. We find that the number of papers cited at least as well as those appearing in high-impact factor journals vastly exceeds the number of papers published in such venues. At the investigator level, this phenomenon extends across gender, racial, and career stage groupings of scientists. We also find that approximately half of researchers never publish in a venue with an impact factor above 15, which, under journal-level evaluation regimes, may exclude them from consideration for opportunities. Many of these researchers publish equally influential work; however, raising the possibility that the traditionally chosen journal-level measures that are routinely considered under decision-making norms, policy, or law, may recognize as little as 10%–20% of this influential work.

## Introduction

Biomedical hiring and promotion committees use journal-level heuristics because the quantity of documentation that researchers provide in their applications exceeds the attentional capacity that reviewers can provide [1]. McKiernan and colleagues [2] noted in their analysis of review, promotion, and tenure (RPT) guidelines that 63% of their sample of institutions that mention journal citation rates associate this with

**Data availability statement:** Article-level data used in this paper are available at Figshare (https://doi.org/10.35092/yhjc.c.4586573). Anonymized author-level derivative data and code used are available at Figshare (https://doi.org/10.6084/m9.figshare.24216822).

**Funding:** Support for this work was provided by the Office of the Vice Chancellor for Research and Graduate Education at the University of Wisconsin–Madison to BIH (https://www.wisc.edu) with funding from the Wisconsin Alumni Research Foundation to BIH (https://www.warf.org/) and the Department of Defense (W911NF2210294 to BIH, https://www.war.gov). The funders had no role in study design, data collection and analysis, decision to publish, or preparation of the manuscript.

**Competing interests:** The authors have declared that no competing interests exist.

**Abbreviations:** ACR, article citation rate; NIH, National Institutes of Health; NCBI, National Center for Biotechnology Information; RCR, Relative Citation Ratio; RPT, review, promotion, and tenure.

quality. They conclude that "these documents show a focus on publication venue and use the JIF [journal impact factor] as a proxy measure for determining how much individual publications should count in evaluations based on where they are published." Ideally, scientific experts would read and apply their scientific expertise to each article under consideration for a given decision about hiring, promotion, or resource allocation [1,3,4]. In some settings, this ideal may be possible; however, in many settings, it is not feasible. For example, tenure-track job advertisements often attract hundreds of applicants, each with dozens of relevant publications to assess. In such a case, the biomedical research community recommended the inclusion of quantitative indicators, but ones that reflect information specific to the articles under consideration, not the venues in which they appear [1,4].

One sensibility that often resonates with members of the scientific community is that if we recognize papers during personnel advancement based in large part on the citation rate of their venues, we should also recognize papers cited equally well, even if appearing in less prestigious venues [5]. In other words, if we recognize a paper published in Cell, we should also recognize papers appearing in journals with lower impact factors that are equally well-cited and valued by practitioners. It should be noted that this view is not ubiquitously shared. However, journal-level metrics fail to identify up to 90% of the equally well-cited papers in the biomedical research literature, controlling for field and year of publication [6]. In other words, the sensitivity of the biomedical community's measure of meritorious papers via their journals' citation rate is on the order of 10%.

Here, we measure the degree of potential misallocation of recognition based on the continuing use of journal-level metrics at the level of individual articles and investigators. Notably, we use a common approach, impact factor thresholding [7], to identify papers published in prestigious venues ("Journal Elite" papers) as well as those that are equally well-cited but published in any journal ("Citation Elite" papers). We use a journal citation rate (hereafter referred to as "impact factor") of 15 citations per article per year as our main threshold, but test 10 and 20 citations per paper per year as well for robustness checks. We find that the number of papers that would receive more recognition using article-level metrics instead of journal-level metrics is several-fold. Notably, half of biomedical researchers have never published in an elite journal, but a large fraction of these have published equally well-cited papers.

## Results

### Journal-level metrics overlook the most influential science

In a previous study, we asked how many highly influential papers are actually published in the most prestigious journals, such as *Science*, *Nature*, and *Cell* [6], using the Relative Citation Ratio (RCR), an article-level citation indicator that accounts for field and year of publication [5,6,8–10]. We showed that there are nearly 10 times more papers that achieve citation levels comparable to those in prestigious venues such as *Cell*, *Nature*, and *Science*, yet appear in less prestigious venues. This builds on previous work showing that there has been a decreasing correlation over the decades between journal impact factors and individual article citation rates (ACRs)

PLOS Biology

[11]. Number of publications can also influence career advancement [12]. However, because we fixed the number of publications and compared only the article-versus journal-level metrics for each paper, this is controlled for in our analysis. There is a weak correlation of 0.08 between the number of an author's publications per year and citation statistics, but it may not rise to the level of practical significance (see Supporting information). These results raise the alarming possibility that, in identifying influential papers exclusively based on the prestige of the journal in which they appear, research assessment now overlooks the vast majority of equally influential articles. However, the extent of this oversight may vary depending on the precise journal citation rate threshold chosen to identify "prestigious" journals.

To test the robustness of our findings, we varied the impact factor threshold used to classify a journal as "prestigious" or not. We determined the median RCR of papers published in journals matching or exceeding the chosen impact factor threshold (impact factor ≥ 15, "Journal Elites"). Articles with an RCR higher than the median of those published in journals above the impact factor thresholds were labeled as article-level "Citation Elites" (Fig 1A and 1B). Throughout this paper, we assess the robustness of findings to different choices of impact factor thresholds and find them to be generally invariant (S1 Text). We find that most influential papers are published in journals below the impact factor threshold defining "prestigious" (Fig 1C). This result raises a question: where are these other influential papers published? The modal response depends slightly on the impact factor threshold used to define "prestigious," but in general, such highly influential papers are published in journals with lower impact factors (Fig 1C). When testing the lower bounds on thresholds, we found that for Journal Elite papers to outnumber Citation Elite papers, the threshold for "prestigious" journals would need to be as low as an impact factor of 3, which is unlikely [7]. Furthermore, the more restrictive the impact factor threshold one uses, the fewer Citation Elite papers are recognized using journal-level measures (Fig 1D).

## Incorporating citation elite papers into evaluation benefits the vast majority of authors

To evaluate how these article-level results generalize to investigator-level indicators, we turned to publicly available, author-level profiles published by the National Center for Biotechnology Information (NCBI) [13]. Biomedical investigators have the opportunity to generate an opt-in public publication profile that can also be used to populate their most relevant publications on National Institutes of Health (NIH) bio-sketches while they apply for funding. We downloaded 50,422 publicly available publication profiles from NCBI in 2022 that matched NIH-funded investigators, ranging from graduate fellows to late-career investigators (Fig 2). We focused on researchers who published at least one article in recent years (i.e., 2010–2019) to exclude artifacts from investigators who left the biomedical research workforce prior to 2010.

The use of journal-level metrics is historically pervasive and entrenched in biomedical research hiring and promotion decisions [1,3]. How might this change if the community also recognized Citation Elite papers? We measured for each investigator the proportion of their set of their primary research articles (e.g., reviews were excluded) that fell into the categories of Citation Elite or Journal Elite. The fraction of scientists that have a higher proportion of Citation Elite papers versus Journal Elite papers is nearly an order of magnitude higher (Fig 2). This is not due to a strong correlation between number of publications and citation rates, which, while statistically significant, was a weak correlation of 0.08 (see Supporting information). This could be viewed as a structural property of the much higher prevalence of highly cited papers compared to the number of articles published per year in high-impact-factor journals. This suggests a substantial improvement in recognition for a large segment of the biomedical research workforce by including article-level indicators as a way of recognizing research.

## Who benefits from article-level recognition?

Structural barriers to equality in the biomedical research workforce have traditionally favored scientists who are more senior, white, or men [14–16]. We asked whether historically underrepresented groups likewise receive more recognition using article- versus journal-level metrics. We therefore examined the distribution of demographic characteristics of those scientists who receive more recognition using journals versus articles (see Methods). For each author, we collated the

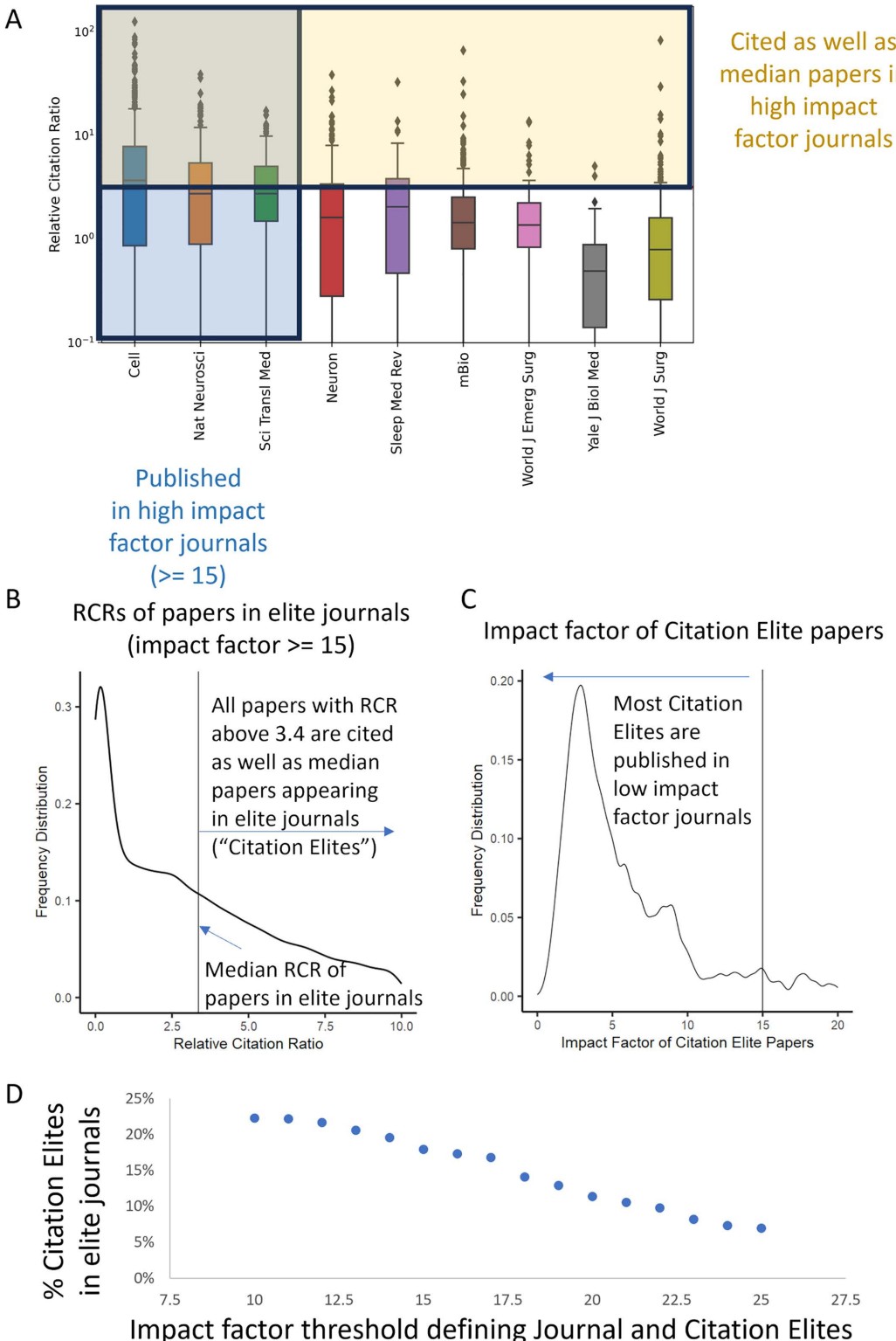

**Fig 1. Most influential papers are published in lower-impact-factor journals. (A)** Schematic for determining which papers are as highly cited as those in prestigious journals. Box and whisker graphs of the Relative Citation Ratio (RCR) of papers from three journals above the selected impact factor threshold are shown (left, blue shading, "Journal Elites") alongside similar graphs for journals with subthreshold (<15 citations per paper per year) are

shown to the right. Articles with RCRs above the line (yellow shading) are considered Citation Elites. All of these journals published multiple Citation Elites. **(B)** Distribution of article-level Relative Citation Ratios for papers published in elite journals (impact factor ≥ 15). **(C)** Impact factor of journals where Citation Elite papers were published. Most papers that are as well cited as those in impact factor ≥ 15 journals are, in fact, published in lower-impact factor journals. **(D)** Generalization of the findings in (C) across different impact factor thresholds. The more stringent the threshold to define an elite journal, the more Citation Elites are published among lower-impact-factor journals. The data underlying this Figure can be found in S1 Data.

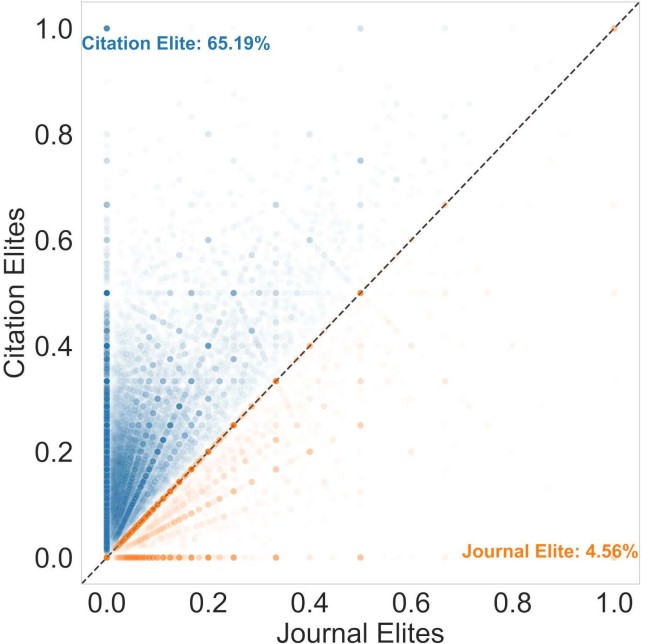

**Fig 2. Scatter plot of the fraction of each researcher's publications (each dot is one researcher) that are above the journal impact factor threshold (x-axis) vs. those that are above the corresponding article-level citation threshold (y-axis).** Researchers above the line would receive more recognition using article-level metrics (hereafter labeled as Citation Elite) while those below the line would receive more recognition using journal-level metrics (hereafter referred to as Journal Elite). Numbers do not add to 100% because some scientists publish no papers that would fall into the Journal Elite or Citation Elite categories. Dots are made semitransparent so that the density of papers occupying a given point is visible with darker shading. The data underlying this Figure can be found in S2 Data.

fraction of papers in their entire profile that met the definition of citation elite or comparatively journal elite. Because the number of papers in each author's profile is fixed between the two measures, the number of publications is controlled for. Overall, 65.19% of researchers would receive more recognition using article-level metrics, while a vastly smaller 4.56% would using journal-level metrics. Of those who receive more recognition with journal-level metrics, 69.53% are men, while 30.47% are women (Fig 3). By contrast, of those who would receive more recognition with article-level metrics 62.28% were men while 37.71% were women (Fig 3), much closer to the overall distribution of men and women in this dataset (38.87% female) (Fig 3). By comparing metrics for men and women in absolute, rather than relative terms, we observe that 18.43-fold more women and 13.34-fold more men receive more recognition with article-level metrics (Fig 3 and Table F in S1 Text).

We next examined the same effects stratified by race. The vast majority of scientists, regardless of race, would receive more recognition using article-level metrics (Fig 3). Statistical analysis confirmed that this is a highly significant effect (Binomial *p*-value < 0.0001, Table A in S1 Text) for each racial category (Asian, Black, White, and Hispanic). The phenomenon of receiving more recognition with article-level measures appears to be broadly shared across racial groups. Recent

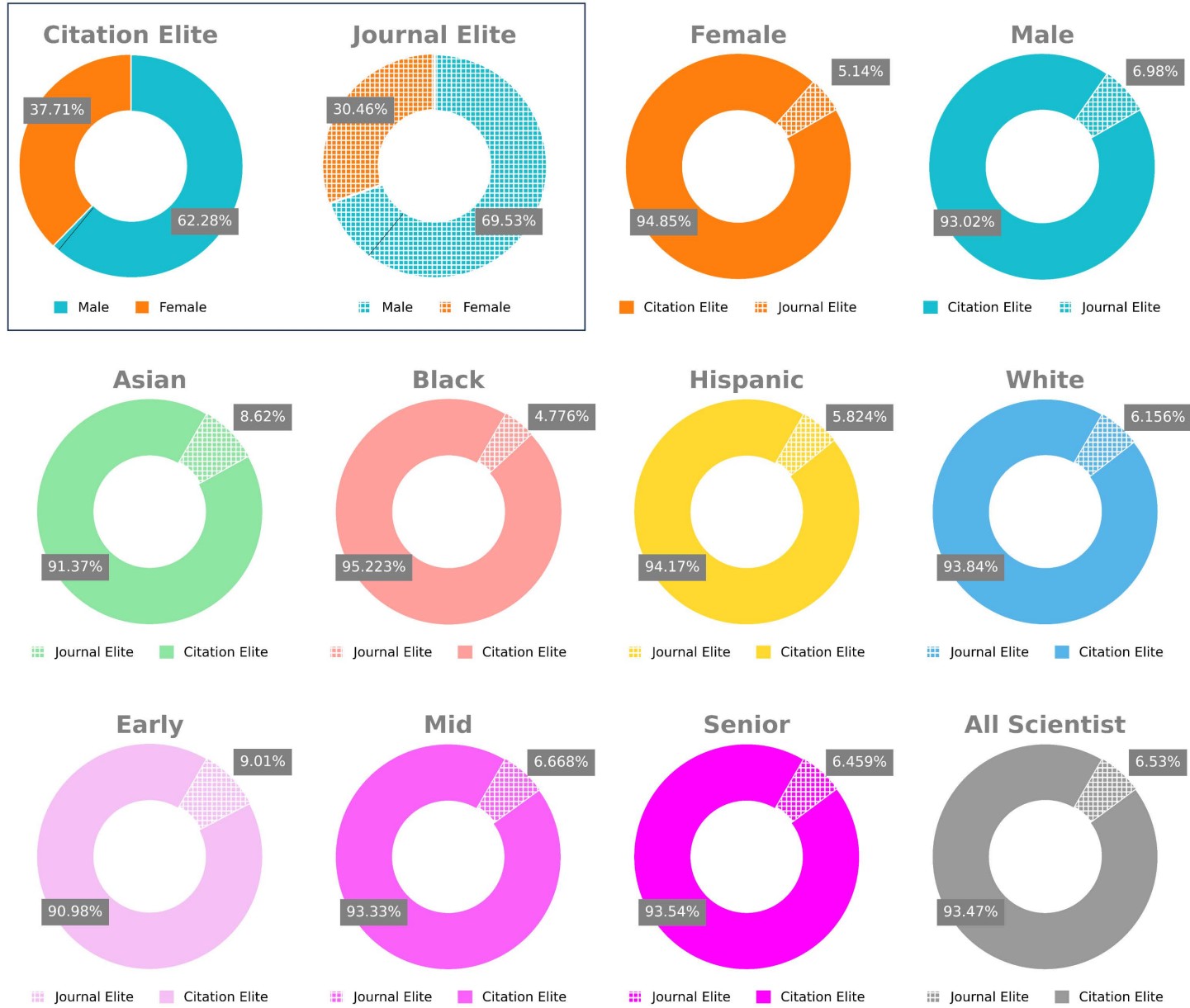

**Fig 3. Demographic analysis of scientists who receive more recognition with journal- vs. article-level measures (RCR).** (Top, within box) Breakdown of how many scientists filtered by those who have more Citation Elite (these scientists are labeled as Citation Elites) papers than Journal Elite papers (labeled as Journal Elites) are male vs. female. Black line, the actual number of male vs. female scientists in this dataset. (Top, outside box) Breakdown of how many scientists filtered by gender receive more recognition with article-level citations (Citation Elite) vs. journal impact factor (Journal Elite). (Middle) Breakdown filtered by race (Binomial $p$-value < 0.0001 across all racial groups). (Bottom) Breakdown filtered by seniority. The data underlying this Figure can be found in S3 Data.

work suggests that applying thresholding to imputed racial predictions as we do here could, in principle, amplify class imbalances inherent in the data [17–19].

We, therefore, confirmed our results on continuous racial prediction scores and found similar results (Fig A in S1 Text). Accordingly, if raw ACRs are used as a substitute for RCR as a robustness check, nearly identical results are observed

(Fig B in S1 Text). These results hold even when subsetting to those top 10,000 researchers ranked by either sorting method, article- or journal-level metrics (see Supporting information).

Finally, we examined the relationship of career stage and the differences between researcher recognition with journal- and article-level metrics (Fig 3 and Table G in S1 Text). As we observed with racial demographics, improvements in recognition are broadly shared across early, mid, and senior career stages.

## Reexamination under strict zero-sum conditions

If article-level measures were considered instead or in addition to journal-level measures, would any career advancement benefit be gained? After all, competition for resources that enable research is often framed as zero-sum, whereby researchers are ranked and compared to one another based on their relative rankings rather than by raw scores. Therefore, we examined not the raw scores that represent the fraction of each author's papers that are labeled as Citation Elite or Journal Elite, as in Figs 2 and 3, but instead their percentile ranking. In other words, each author received a score representing at what percentile their set of publications ranked with R's rank() function for each of these two measures, where 0 was the lowest possible rank and 1 the highest. This would allow a comparison of which authors might be favored under each system under this zero-sum line of thinking. Because this ranking approach places continuous data on a uniform distribution ranging from 0 to 1, it would be expected that for each author that might be favored under journal-level ranked metrics, there would be one other instead favored under article-level metrics. Surprisingly, we observed that this is not the case. Instead, large differences between how many authors are favored using article-level metrics rather than journal-level metrics. Many more researchers published more rank-ordered Citation Elite papers than published rank-ordered Journal Elite papers (47.1% versus 27.8%), while in principle, rank-ordering these measures should equalize these values.

To investigate this phenomenon further, we examined the probability distributions of both raw and rank-ordered statistical distributions. (Fig 4). While the distribution raw scores (fraction of papers labeled as either Citation- or Journal-Elite) are skewed for both journal-level and article-level metrics (Fig 4A and 4B), the Citation Elite frequency distribution has a fatter tail. This indicates more researchers publish such highly cited papers—consistent with the previous results. Rank-ordering the data should produce uniform statistical distributions, aligning with a zero-sum competition framework (e.g., one author at the 95th percentile for each measure). For much of the probability distribution (Fig 4C and 4D), we see that this is the case. However, in both datasets, we observe that there is zero inflation. In other words, for each indicator, some researchers never publish Journal or Citation Elite papers. However, the degree of zero-inflation differs greatly between Journal and Citation Elites (Fig 4C and 4D): More scientists have never published a single journal elite paper than have never published a citation elite paper.

Even using rank-order data, we observe that, in general, researchers are much more frequently recognized using article-level indicators (Fig 5). We observe a similar gender difference favoring article-level metrics using the rank-ordered data as we observed using the continuous raw data. Likewise, we observed that each gender, racial group, and career stage in general would receive more recognition with article-level indicators (Fig 5). Thus, even under a nominally zero-sum framework, diverse researchers would receive more recognition using article-level indicators. A large minority of researchers, who would be overlooked by journal-level metrics because none of their papers were published in a journal above the impact factor threshold, would gain recognition if article-level metrics were used (Fig 6). Therefore, the scarcity of recognition in a journal-level system impacts a sizeable minority of biomedical researchers.

## Discussion

This study investigated the degree of potential misallocation of recognition based on the exclusive use of journal-level metrics in assessing individual investigators, contrasting it with the use of article-level indicators as recommended by the scientific community [1,4]. We find that, using a variety of impact factor thresholds, researchers overwhelmingly receive more recognition with article-level indicators. This advantage cuts across racial, gender, and career groups. Even under

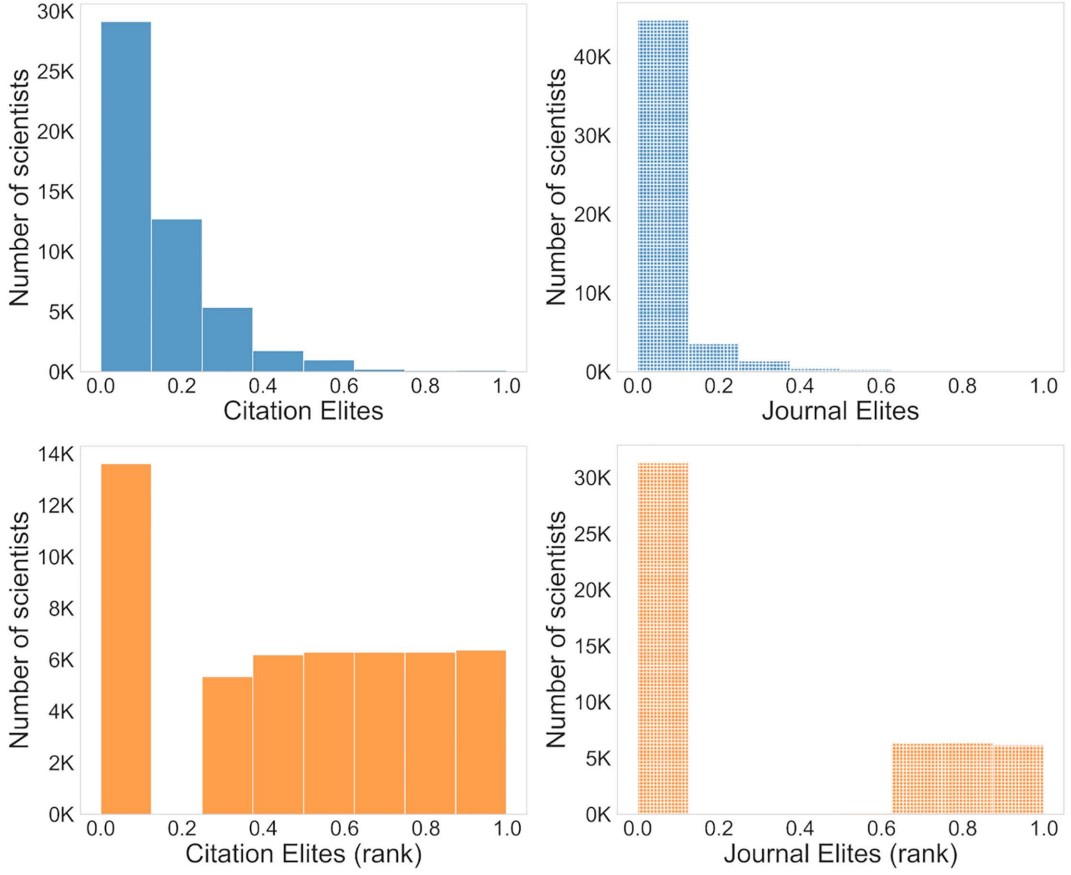

**Fig 4. Distributions of researcher recognition. (A, B)** Frequency distributions of the fraction of each researcher's publications that are recognized under article-level (A, "Citation Elites") vs. journal-level metrics (B, "Journal Elites"). **(C, D)** Frequency distributions after rank-ordering to align the scores with a zero-sum framework using article-level (C) vs. journal-level metrics **(D)**. In principle, rank-ordering should flatten these into uniform distributions, but there is a high degree of zero-inflation. Approximately half of researchers have zero publications in prestigious journals (D, left). By contrast, only a quarter of researchers do not have articles that are cited as well as those published in prestigious journals (C, left). The data underlying this Figure can be found in S4 Data.

zero-sum rank-ordered conditions, we observe that many more researchers would receive recognition using article-level indicators. This is because most researchers never publish in a "prestigious" journal as operationalized by impact factor thresholds, but many published papers in lower impact factor journals are as highly cited as those appearing in these journals.

In 2013, signatories to the San Francisco Declaration on Research Assessment acknowledged the pervasive harm that the continued use of journal-level heuristics in funding, hiring, and promotion decisions causes to scientists [1,4]. This mental heuristic is akin to judging a paper by the company it keeps (the average citation rate of all the other papers published in the journal) rather than by the article's own downstream influence. Signatories acknowledged that, while the use of scientific judgment about a paper is the gold standard, in many decision-making contexts where there are potentially thousands or millions of scientific articles to compare, it is necessary to augment expert judgment with metrics. These should be article-level to ensure that the information is specific to the article(s) in question, rather than by the company they keep.

Our results echo and underscore the importance of considering article-level metrics in assessing scholars and scholarships. While citations are a form of recognition by the community, journal names are more visible to reviewers, who often

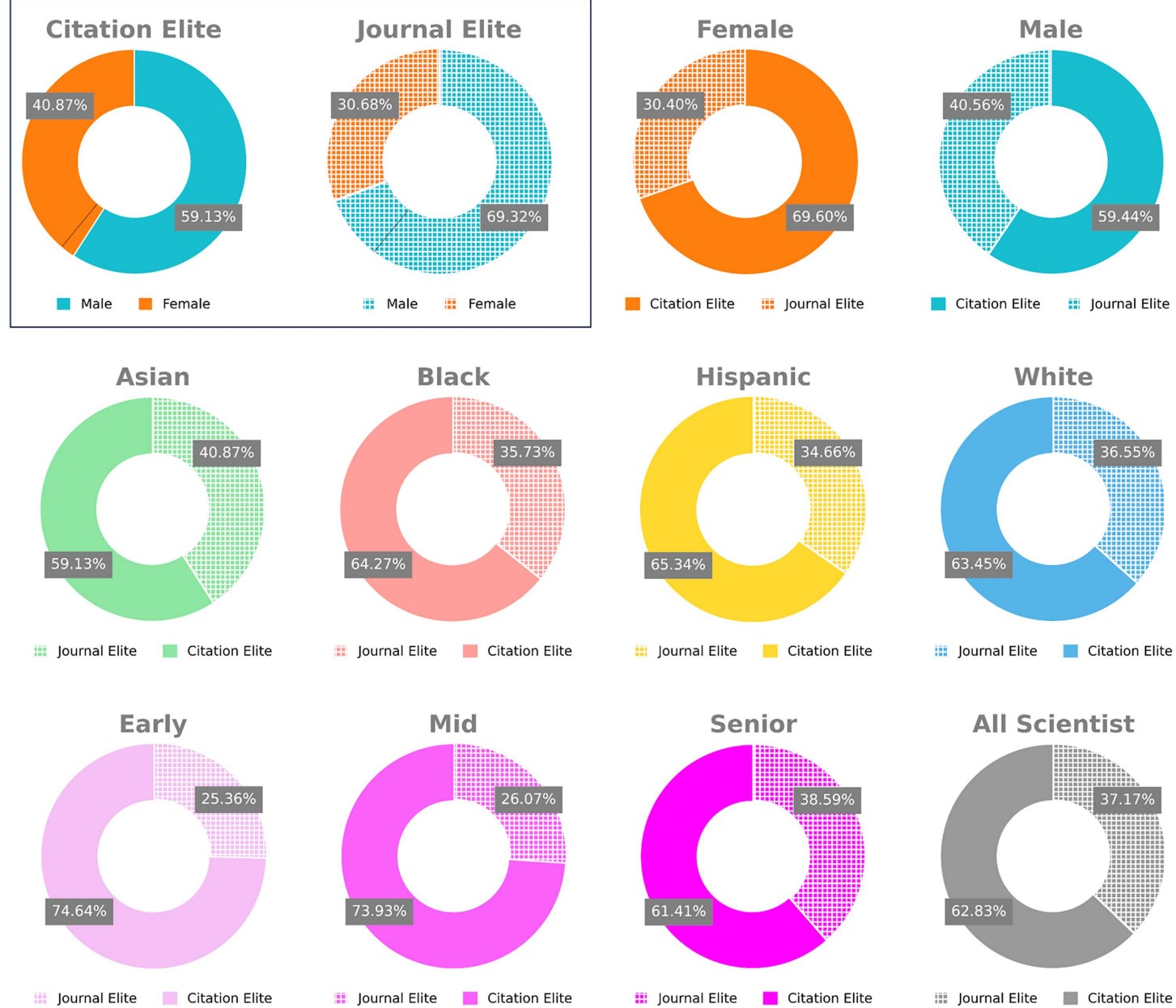

**Fig 5. Demographic analysis of scientists who receive more recognition with journal- vs. article-level (RCR) measures using zero-sum rankings.** (Top) Breakdown by gender. (Middle) Breakdown by race. (Bottom) Breakdown by seniority. The data underlying this Figure can be found in S5 Data.

use journal-level metrics such as journal impact factors as a shorthand for scientific impact. To assess article-level impact, reviewers would need to investigate each paper's citation record, which is rarely done in practice unless these metrics are included in curricula vitae [5]. This is not to diminish by any means the value of journal-level metrics for assessing journals (rather than papers) or as additional information in combination with article-level metrics [20], but to underscore that article-level metrics provide important insights, especially for researchers who publish impactful work outside of

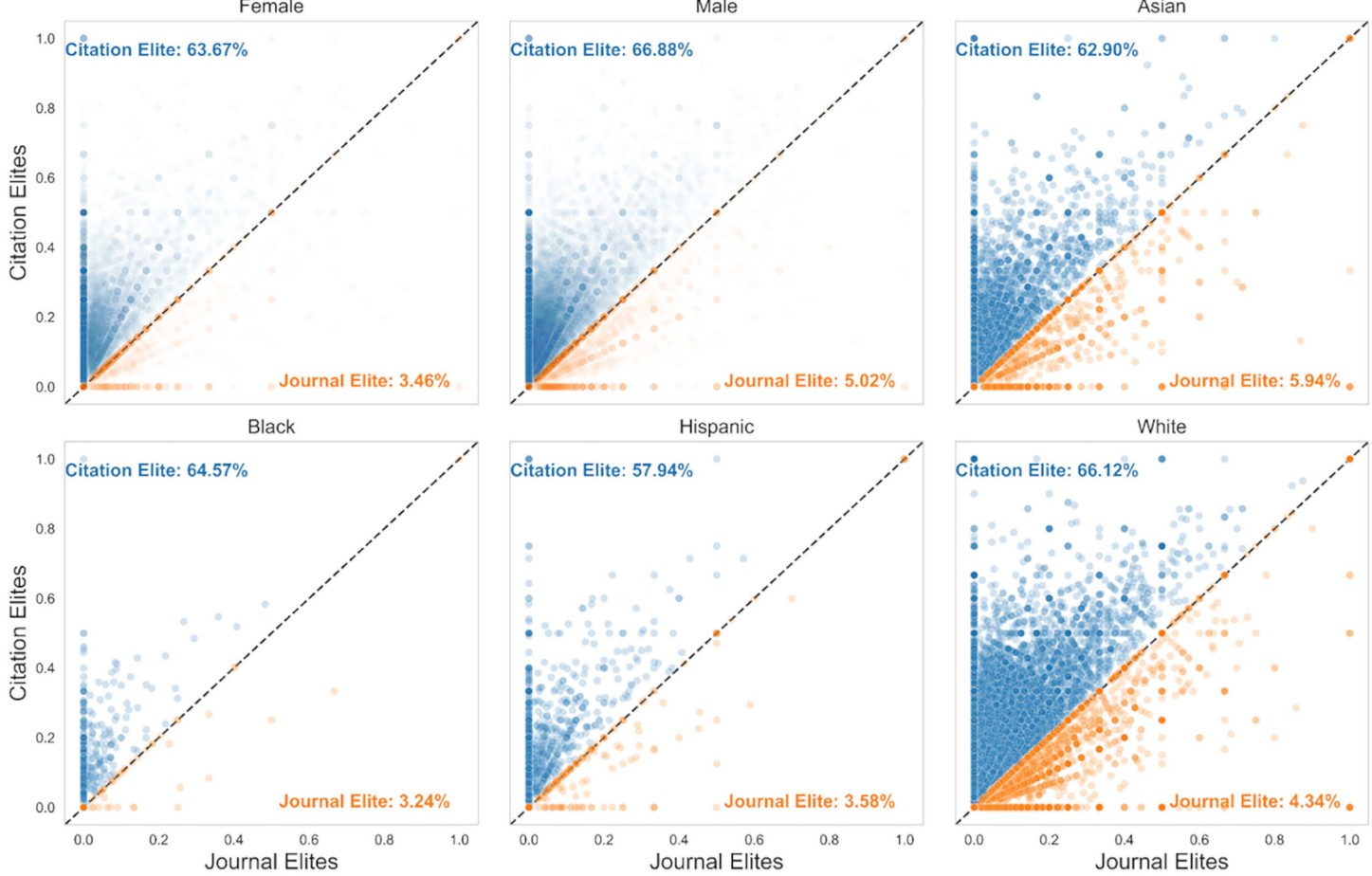

**Fig 6. Scatter plots of the fraction of researchers' papers are Journal Elite (x-axis) vs. Citation Elite (y-axis), stratified by gender or race.** The data underlying this Figure can be found in S6 Data.

prestigious venues for various reasons. While not explored here, other measures of a scientific article, such as the publication of clinically relevant trials or guidelines, are not explored in this study, but such article-level measures have been developed and make a valuable addition [21,22].

A dynamic, sustainable, and diverse scientific workforce is vital for a country's economic growth, global competitiveness, and ability to address pressing societal challenges. Yet, the sustainability of the U.S. biomedical research workforce has been an ongoing concern, particularly given its skewness towards older researchers [15,23], and underrepresentation of African American/Black researchers [16,24,25]. Furthermore, the U.S. scientific workforce shows historically male-dominated structure [26–28]. The gender disparities may be attributed to various factors, among which is female researchers' limited access to resources and opportunities for publishing in esteemed journals [26,28–30]. Consequently, the barriers faced by women in publishing in these journals result in unfavorable evaluations based on journal-level metrics. Given the consequences of such evaluation, female researchers would be further disadvantaged, as suggested by our results. Other underrepresented groups in the biomedical scientific workforce would also suffer for similar reasons.

Such trends exacerbate worries about the workforce's future. These topics have been discussed at congressional appropriations hearings [31], motivated specific appropriations for the NIH in the 21st Century Cures Act [15], and have been the subject of written testimony from NIH leadership to Congress [32]. NIH Initiatives, such as the Next Generation

Research Initiative and Faculty Institutional Recruitment for Sustainable Transformation initiative, have attempted to address some of these workforce problems [32,33]. However, the scientific community itself bears responsibility for structural issues that persist. The inappropriate use of journal-level metrics, as highlighted by DORA [1], remains a key challenge, as it amplifies recognition for established, often white, male researchers while marginalizing others [24,25], potentially amplifying structural inequalities in the workforce. Results from this study offer further evidence that the utilization of article-level metrics may contribute to addressing the broader structural imbalance in our scientific workforce and help build a competitive scientific workforce.

Despite widespread criticism [1,4,7], journal-level metric such as journal impact factor remains a key criterion in RPT decisions in many institutions [34–36]. Several studies reported its inclusion as an important criterion for hiring, tenure, performance evaluation, and promotion, and this trend persists across countries and faculty positions [35,37], despite being well-critiqued as an evaluator of the quality of publications [38]. Additionally, a decent percentage of research-intensive as well as Master's institutions refer to journal impact factor or related terms and encourage their usage in RPT documents [47]. Moreover, this metric is often considered correlated with the quality, significance, and impact of research by the institutions and review boards. Therefore, many researchers in the biomedical community feel that they are unfairly judged in research assessment, arguing that they do not receive enough credit given the influence on the research community of their published work [39]. Our analysis supports this sentiment, showing that a significantly larger number of researchers, particularly women and underrepresented groups, would gain greater recognition through article-level metrics.

In conclusion, the current system, which prioritizes journal prestige, systematically overlooks the contributions of many influential researchers. The artificial scarcity of recognition enforced by journal prestige not only diminishes the visibility of influential work but systematically excludes a large number of highly impactful scientists from receiving the recognition they deserve. Our study highlights the need for broader adoption of article-level metrics to foster a more equitable and accurate recognition system in science in order to build a competitive scientific workforce.

It might be argued that improvements in recognition are impossible under the zero-sum conditions of the biomedical research enterprise, since the number of positions or honors is fixed, any gain for one researcher necessarily entails a loss for another. There are two rejoinders to this line of argumentation, one direct and one indirect. The first is that additional information can still improve the fairness and accuracy of recognition, even in zero-sum systems. Researchers who have no Journal Elite papers but do have Citation Elite papers would benefit from the acknowledgement that their work, despite being published in less elite journals, has had comparable influence to that in top-tier venues. This may indeed improve their relative ranking and decrease that of others. But for it to be true that no improvement in recognition has been achieved, one would have to assert that even highly cited papers in lower-tier journals are undeserving of recognition solely because of their publication venue. The existence of a zero-sum system does not obviate the reduction of uncertainty in decision-making that an additional, or even alternative, information source provides. In fact, it may be in zero-sum hypercompetitive conditions that such improvements to the decision-making process are most needed.

The second rejoinder to this argument is that while it may be true that the use of article-level measures does not increase the number of positions at a given moment, they can contribute to expanding opportunities over time. We have powerful examples of article-level metrics contributing to growth in overall biomedical research funding by engendering trust with legislators who set the budget. Former NIH Director Francis Collins used article-level metrics as the building blocks of bird's-eye-view analyses of the biomedical funding landscape to continuously evaluate the research portfolio, make difficult policy decisions, and demonstrate effectiveness to the United States legislature during appropriations meetings [31]: "These types of tools provide NIH with the evidence base it needs to make funding decisions that promote an efficient and impactful biomedical research portfolio." Initiatives to increase the overall research budget, as seen for NIH from 2016 to the pandemic, are more credible when accompanied by rigorous quantification that starts from a granular level and aggregates up to a landscape view. Demonstrating that scientists can quantify and even forecast

[8,16,21,23,32,40–44] outcomes of the research enterprise beyond what peer review alone provides represents an important step towards sustaining and expanding the biomedical research enterprise. Because trust in science has declined since the public health policies during the pandemic, every source of added confidence is needed to foster a growing research enterprise.

Like many other studies, our research is not immune to limitations. One caveat to this work is that we do not examine the extent to which past influence predicts future influence, though research along these lines has already been conducted [6]. Neither do we explore the dynamics of authors purchasing fraudulent citations or authorship in journals. Furthermore, while we focus on citation impact of scholars and scholarships, the ultimate beneficiaries of biomedical research are patients, and there is growing interest in evaluating research based on societal outcomes [8,21,22,43,45,46]. Future research could explore how article-level measures align with societal impact, particularly in clinical settings.

## Methods

### Publication profiles

Publicly available data on funded NIH grants were downloaded in bulk from NIH ExPORTER [47]. From these, names of principal investigators (who could be graduate students on training grants through senior investigators funded on long-term research project grants) were extracted. Scientists who lacked either an NIH grant or a public MyNCBI profile were considered out of scope. From these names, those investigators who opted in to posting a public publication profile on MyNCBI [13] were crawled using the MyNCBI URL structure that incorporates the name representation found in NIH ExPORTER data. PubMed [48] identifier numbers for Medline papers were extracted from those public sites to construct a profile of each researcher's publications. The publicly available WRU [49] and gendeR [50] R packages were used for race and gender imputation. We define the early career researchers as those who have a career age of five years or less, and mid-career researchers as those who have a career age of 10 years or less but more than five [51]. The rest are categorized as senior researchers.

### Identifying prestigious articles using journal- and article-level metrics

Biomedicine has long recognized papers published in prestigious venues as influential. Many have advanced the view that papers that are as highly cited as those appearing in such prestigious journals should also be recognized as influential. We used a common heuristic, impact factor thresholding [6], to identify papers published in prestigious journals (i.e., using an impact factor threshold of 15, papers published in journals with an impact factor of 15 or more would be recognized as influential in a journal-level recognition regime). We used the journal citation rate in the iCite database as our measurement of journal impact factors [6,9,21,52–55].

Under the sensibility that papers that are as highly cited as those appearing in such prestigious journals should also be recognized as influential, we used the NIH RCR [6,9], which measures influence as the number of citations per year an article has received, accounting for its field of research and publication year. Only articles tagged as research articles, which exclude article types such as reviews and editorials, were considered. It should be noted that the use of RCR has stirred debate [9,56,57]. For this reason, we also measured Citation Elites in terms of the (ACR, total citations per year elapsed since publication).

For a given impact factor threshold, articles were flagged as influential using journal-level metrics ("Journal Elites") if they were published in a journal whose impact factor exceeded the chosen threshold in the year of the article's publication.

For article-level metrics, the following procedure was used to identify papers recognized as influential ("Citation Elites") as those in a prestigious journal selected by the given impact factor threshold:

1. Identify papers published in a journal whose impact factor exceeded the chosen threshold

2. Measure the median RCR (or ACR) of papers published in these high-impact-factor journals ($RCR_{med}$ or $ACR_{med}$)

3. Flag as influential any paper exceeding $RCR_{med}$ or $ACR_{med}$

The same procedure was used for analyses using citations/year, substituting that measure for RCR as a robustness check (see Fig B in S1 Text). Our definition of citation elites is not without limitations. The theoretical upper bounds are different for precision (when a paper is flagged as a journal elite, it is also found to be a citation elite) versus recall (when a paper is flagged as a citation elite, it is also found to be a journal elite). The theoretical upper bound for recall under this definition is 100% (all citation elites are found in journals above the specified impact factor threshold). The theoretical upper bound for precision is, by definition 50%, since the median RCR or ACR is used for defining a citation rate threshold. Here, recall is the accuracy measure that is investigated because it is responsive to our research questions (e.g., how many of the citation elites are actually identified by journal impact factor). For percentile ranks, the percent_rank() function from dplyr 1.1.1 was used in R 4.2.2.

Notably, this definition of what constitutes a Citation Elite has limitations. The first is that some might consider it to be favored for detecting Citation Elite papers in high-impact factor journals (above the Journal Elite threshold, i.e., 10, 15, or 20 citations per article per year) rather than in low-impact factor journals (below Journal Elite threshold). Because the Citation Elite threshold is set to the median RCR or ACR of Journal Elite papers, 50% of papers in such journals will always be flagged as Citation Elites. However, this definition does not control the number of Citation Elites in lower-impact factor journals. There is no a priori reason why any papers published in less prestigious journals should be cited as frequently as those in elite journals. Nevertheless, we observe that the vast majority of Citation Elites do in fact reside in lower-impact factor journals. In addition, the presence of Citation Elites in non-elite journals does not influence either the Citation Elite status or the Journal Elite status of papers in journals above the impact factor threshold. These would be flagged the same even if all Citation Elites in non-elite journals were deleted from the database, in which case all Citation Elites would appear in above-threshold journals.

An earlier version of this manuscript appeared as a preprint [58].

**Data collection**

For this analysis, we pulled 50,422 publicly available author profiles from the NCBI in 2021 that matched NIH-funded investigators. For each author, we extracted the following features to conduct our analysis:

start year(int64): The year with the earliest publication of the author as per the database record

end year(int64): The year with the latest publication of the author as per the database record

decade(int64): The decade of the earliest publication of the author

article level metric(float64): fraction of papers of the author that has RCR equal to or higher than the median RCR of papers published above the specific JIF threshold

journal level metric(float64): fraction of papers of the author that meet or exceed the specific impact factor threshold

article level metric_rank(float64): percentile rank of article level metric

journal level metric_rank(float64): percentile rank of journal level metric

rcr_median(float64): median rcr of the papers with

appl(int64): application id associated with the name from the database

surname(object): the surname of the author

forename(object): the forename of the author

forename_longest(object): for multiple forenames, the one with the most characters

male(float64): proportion of male names based on U.S. census data with the forename_longest of the author using R's gendeR package

female(float64): proportion of female names based on U.S. census data with the forename_longest of the author using R's gendeR package

gender(categorical): defined based on the largest of male and female

white(float64): the probability of being white inferred from surname using R's wru package

black(float64): the probability of being black inferred from surname using R's wru package

his(float64): the probability of being Hispanic inferred from surname using R's wru package

asi(float64): the probability of being Asian inferred from surname using R's wru package

oth(float64): the probability of being anything other than white, black, hispanic, and asian inferred from surname using R's wru package

gender(categorical): defined based on the largest of the race probabilities

art_better(bool): Binary class for whether a researcher appears more competitive under article level metric compared to journal level metric

journ_better(bool): Binary class for whether a researcher appears more competitive under journal level metric compared to article level metric

art_journ_same(bool): Binary class for whether journal level metric equals to article level metric

## Supporting information

**S1 Text. Supplemental materials. Table A**: Binomial test on race: We conducted 2-sided binomial test and used binom_test function in the stats module from SciPy library and Python 3.8.8 version for the binomial test. All 4 racial categories showed significance across all three thresholds (impact factor $\geq$ 10, 15, or 20, respectively). **Table B**: Chi-squared test on gender (all 3 thresholds showed significance): We used Python's stats module from SciPy library and python 3.8.8 version for the Chi-squared test. **Table C**: Chi-squared test on race: We used Python's stats module from SciPy library and Python 3.8.8 version for the Chi-squared test. We classified each scientist in a particular race if the probability of the researcher being in that racial group is greater than 0.50. **Table D**: Paired sample $t$ test on Citation Elites versus Journal Elites based on career stage: We conducted a matched-pairs $t$ test for each career stage across all the thresholds. We used Python's stats module from the SciPy library and the Python 3.8.8 version for the test. We define early career researchers as those who have a career age of 5 years or less, and mid-career researchers as those who have a career age of 10 years or less but more than five. The rest are categorized as senior researchers. **Table E**. Investigators stratified by race whose papers are most frequently recognized in the Journal Elite versus Citation Elite categories. Chi-squared table for race at an impact factor threshold of 15. Chi-squared: 50.026, $p$-value: 7.887060421743808e-11. **Table F**. Investigators stratified by gender whose papers are most frequently recognized in the Journal Elite versus Citation Elite categories. Chi-squared: 40.575, $p$-value: 1.8922670956226282e-10′. **Table G**: Investigators stratified by career stage whose papers are most frequently recognized in the Journal Elite versus Citation Elite categories. Chi-squared: 8.289, $p$-value: 0.0158. **Table H:** KS Test on Racial Probability: We conducted 2-sided k-s test on each racial probability score for scientists who

receive more recognition under article versus journal-level metrics (also see Fig A in S1 Text). The p-value of the k-s test is significant across all three thresholds (impact factor ≥ 10, 15, or 20, respectively). **Table I**: Proportion-z test on race: We conducted 2-sided proportion z test across all racial categories using proportions_ztest function in the stats module from statsmodels library and Python 3.8.8 version for the test. We get similar significance across all thresholds. **Table J:** Percentage of male and female scientists in citation elite and journal elite. **Table K:** Percentage of citation elite and journal elite across gender groups. **Table L:** Percentage of citation elite and journal elite across racial groups. **Table M:** Percentage of citation elite and journal elite across career groups. **Table N:** Percentage of citation elite and journal elite across the entire dataset. **Table O:** Statistical tests on threshold 15. **Table P:** Statistical tests on threshold 10. **Table Q:** Statistical tests on threshold 20. **Fig A**: Cumulative probability distribution of racial score of scientists who are more recognized as Citation Elites versus Journal Elites. p-values shown in Table H in S1 Text. The data underlying this Figure can be found in S7 Data. **Fig B**: Demographic analysis of scientists who receive more recognition with journal- versus article-level measures (Article citation rate). (Top, within box) Breakdown of how many scientists filtered by those who have more Citation Elite (these scientists are labeled as Citation Elites) papers than Journal Elite papers (labeled as Journal Elites) are male versus female. (Top, outside box) Breakdown of how many scientists filtered by gender receive more recognition with article-level citations (Citation Elite) versus journal impact factor (Journal Elite). (Middle) Breakdown filtered by race. (Bottom) Breakdown filtered by seniority. The data underlying this Figure can be found in S8 Data.
(DOCX)

**S1 Data.  Data underlying Fig 1.**
(XLSX)

**S2 Data.  Data underlying Fig 2.**
(XLSX)

**S3 Data.  Data underlying Fig 3.**
(XLSX)

**S4 Data.  Data underlying Fig 4.**
(XLSX)

**S5 Data.  Data underlying Fig 5.**
(XLSX)

**S6 Data.  Data underlying Fig 6.**
(XLSX)

**S7 Data.  Data underlying Fig A in S1 Text.**
(XLSX)

**S8 Data.  Data underlying Fig B in S1 Text.**
(XLSX)

## Author contributions

**Conceptualization:** Chaoqun Ni, B. Ian Hutchins.

**Data curation:** B. Ian Hutchins.

**Formal analysis:** Salsabil Arabi, B. Ian Hutchins.

**Funding acquisition:** B. Ian Hutchins.

**Investigation:** Salsabil Arabi, Chaoqun Ni, B. Ian Hutchins.

**Methodology:** Chaoqun Ni, B. Ian Hutchins.

**Project administration:** B. Ian Hutchins.

**Resources:** B. Ian Hutchins.

**Software:** Salsabil Arabi, B. Ian Hutchins.

**Supervision:** Chaoqun Ni, B. Ian Hutchins.

**Validation:** Salsabil Arabi, Chaoqun Ni, B. Ian Hutchins.

**Visualization:** Salsabil Arabi, B. Ian Hutchins.

**Writing – original draft:** Salsabil Arabi, Chaoqun Ni, B. Ian Hutchins.

**Writing – review & editing:** Salsabil Arabi, Chaoqun Ni, B. Ian Hutchins.

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
