## [Editor Report · Decision Letter 0]

12 Sep 2025

Dear Ian,

Thank you for submitting your revised manuscript entitled "Analysis of citation dynamics reveals that you do not receive enough recognition for your influential science" for consideration as a Meta-Research Article by PLOS Biology.

Your revisions have now been evaluated by the PLOS Biology editorial staff and I'm writing to let you know that we would like to send your submission out for external peer review.

IMPORTANT: I note that you have not supplied a marked-up "track changes" version of your manuscript. Please provide this when you upload your additional metadata (see next paragraph).

Once your full submission is complete, your paper will undergo a series of checks in preparation for re-review. After your manuscript has passed the checks it will be sent out for review. To provide the metadata for your submission, please Login to Editorial Manager (https://www.editorialmanager.com/pbiology) within two working days, i.e. by Sep 16 2025 11:59PM.

Kind regards,

Roli

Roland Roberts, PhD

Senior Editor

PLOS Biology

rroberts@plos.org

---

## [Decision Letter · Decision Letter 1]

17 Oct 2025

Dear Ian,

Thank you for your patience while we considered your revised manuscript "Analysis of citation dynamics reveals that you do not receive enough recognition for your influential science" for consideration as a Meta-Research Article at PLOS Biology. Your revised study has now been evaluated by the PLOS Biology editors, the Academic Editor and the original reviewers.

You'll see that reviewer #1 is satisfied and has no further requests. Reviewer #2 says that “the empirical findings are of significant interest,” but he disagrees substantially with several aspects of your interpretation. He also thinks that you should explicitly flag that the RCR metric’s merits have been questioned.

In light of the reviews, which you will find at the end of this email, we are pleased to offer you the opportunity to address the remaining points from the reviewers in a revision that we anticipate should not take you very long. We will then assess your revised manuscript and your response to the reviewers' comments with our Academic Editor aiming to avoid further rounds of peer-review, although we might need to consult with the reviewers, depending on the nature of the revisions.

**IMPORTANT - SUBMITTING YOUR REVISION**

*Resubmission Checklist*

*Published Peer Review*

*PLOS Data Policy*

Sincerely,

Roli

Roland Roberts, PhD

Senior Editor

PLOS Biology

rroberts@plos.org

REVIEWERS' COMMENTS:

Reviewer #1:

The authors have fully addressed my concerns. I congratulate them to their important work.

Reviewer #2:

[identifies himself as Ludo Waltman]

[IMPORTANT: see the fully formatted version at https://prereview.org/reviews/17335296]

My review is available online at https://prereview.org/reviews/17335296.

This is an interesting paper presenting a large-scale comparison between the use of journal-level and article-level citation indicators for assessing biomedical researchers.

While the empirical findings reported by the authors are of significant interest, I disagree with the interpretation the authors give to their findings. In their interpretation of Figure 2, the authors conclude that their results suggest “a substantial improvement in recognition for a large segment of the biomedical research workforce by including article-level indicators as a way of recognizing research”. I disagree with this interpretation because in most contexts, in particular hiring, promotion, and funding allocation, researchers essentially find themselves in a zero-sum setting. If one researcher gets more recognition (e.g., is more likely to be hired, promoted, or funded), this implies that some other researcher will get less recognition (e.g., will be less likely to be hired, promoted, or funded).

In the final subsection in the Results section, the authors recognize that researchers often find themselves in zero-sum settings. Surprisingly, however, the authors claim that even in a zero-sum setting there are “large differences between how many authors are favored using article level metrics rather than journal level metrics”. This is an odd conclusion. In a zero-sum setting, the number of authors favored using one indicator must by definition be equal to the number of authors favored using some other indicator.

This odd conclusion turns out to follow from the specific statistical approach the authors take (without explaining it in full detail) to convert raw indicator scores into percentile ranks, as illustrated in Figure 4. According to the authors, after converting raw indicator scores into percentile ranks, there are still many more researchers who receive recognition using article-level indicators than researchers who received recognition using journal-level indicators. However, this is an artefact of the statistical approach taken by the author. In the real world, the number of opportunities to be hired, to be promoted, or to be funded is fixed, so in the real world these settings are of a truly zero-sum nature. If one researcher receives more recognition and is therefore more likely to be hired, promoted, or funded, there must be some other researcher who receives less recognition and is less likely to be hired, promoted, or funded. The use of article-level indicators instead of journal-level indicators will not result in an increase in the number of researchers who can be offered a job or a promotion, or who can be awarded funding.

There are important reasons to criticize excessive reliance of journal-level indicators, but the argument presented by the authors is not convincing. The empirical insights provided by the authors are valuable, but the authors need to rethink the interpretation they give to their findings.

Finally, a minor comment I have relates to the RCR indicator used by the authors to measure the citation impact of an article. There has been some debate about this indicator. See the following paper by Janssens and colleagues: https://doi.org/10.1371/journal.pbio.2002536. And see also the following blog post by myself: https://www.cwts.nl/blog?article=n-q2u294. In my view, the authors should inform readers that there are different perspectives on the pros and cons of the RCR indicator.

---

## [Editor Report · Decision Letter 2]

6 Nov 2025

Dear Ian,

Thank you for your patience while we considered your revised manuscript "Analysis of citation dynamics reveals that you do not receive enough recognition for your influential science" for publication as a Meta-Research Article at PLOS Biology. This revised version of your manuscript has been evaluated by the Academic Editor.

Based on the Academic Editor's assessment, we are likely to accept this manuscript for publication, provided you satisfactorily address the following data and other policy-related requests.

IMPORTANT - please attend to the following:

a) Please could you make your Title more explicit for readers who are coming to the topic afresh. We suggest "Analysis of citation dynamics reveals that most researchers would receive more recognition if assessed by article level metrics than by journal level metrics" or "Analysis of citation dynamics reveals that most researchers would be better served if assessed by article level metrics than by journal level metrics"

b) Please address my Data Policy requests below; specifically, we need you to supply the numerical values underlying Figs 1ABCD, 2, 3, 4, 5, 6, S1, S2, either as a supplementary data file or as a permanent DOI’d deposition. I note that you already have associated Figshare depositions, but these seem to contain largely raw data, so please also supply the values directly underlying the Figures.

c) Please cite the location of the data clearly in all relevant main and supplementary Figure legends, e.g. “The data underlying this Figure can be found in S1 Data” or “The data underlying this Figure can be found in https://figshare.com/XXXXXXXX

d) Please make any custom code available, either as a supplementary file or as part of your data deposition.

e) Please include the URLs of your funders in the Financial Disclosure statement.

We expect to receive your revised manuscript within two weeks.

*Published Peer Review History*

*Press*

Sincerely,

Roli

Roland Roberts, PhD

Senior Editor

rroberts@plos.org

PLOS Biology

DATA POLICY:

Regardless of the method selected, please ensure that you provide the individual numerical values that underlie the summary data displayed in the following figure panels as they are essential for readers to assess your analysis and to reproduce it: Figs 1ABCD, 2, 3, 4, 5, 6, S1, S2. NOTE: the numerical data provided should include all replicates AND the way in which the plotted mean and errors were derived (it should not present only the mean/average values).

CODE POLICY

DATA NOT SHOWN?

---

## [Editor Report · Decision Letter 3]

13 Nov 2025

Dear Ian,

Thank you for the submission of your revised Meta-Research Article "An analysis of citation dynamics reveals that most researchers would receive more recognition if assessed by article level metrics than by journal level metrics" for publication in PLOS Biology. On behalf of my colleagues and the Academic Editor, Malcolm Macleod, I'm pleased to say that we can in principle accept your manuscript for publication, provided you address any remaining formatting and reporting issues. These will be detailed in an email you should receive within 2-3 business days from our colleagues in the journal operations team; no action is required from you until then. Please note that we will not be able to formally accept your manuscript and schedule it for publication until you have completed any requested changes.

IMPORTANT: You'll see I've had to flip your Title so that "an analysis of citation dynamics" occurs at the beginning. This is because we try to avoid punctuation (and certainly colons) in Titles. Sorry not to have mentioned this earlier, and hope this is OK.

Sincerely, 

Roli

Senior Editor

PLOS Biology

rroberts@plos.org